# Peer review of "Transforming Clinical Research: The Power of High-Throughput Omics Integration"

_proteomes, 2024, doi:10.3390/proteomes12030025_

Round 1

Reviewer 1 Report

Comments and Suggestions for Authors

The manuscript of Rui Vitorino presents a comprehensive review of the current high-throughput omics pipelines, covering key technologies like proteomics, peptidomics, metabolomics, next generation sequencing and gene expression, protein/DNA/RNA arrays coupled with data integration techniques and their diverse applications in the basic research and the biomedical field. 

In addition, the review presents the critical role of bioinformatics tools, clearly presented in the table 6 and highlights the major statistical methods in managing the large datasets generated by these technologies.

I recommend the review to be published in the Proteomes journal after minor revisions:

1. The graphic abstract and the graph/scheme presented in the Figure 1 should be changed by using Arial font and larger size (for Figure 1 especially which is hard to be analyzed in the format presented in this version of the manuscript).

2.  Please add additional databases like CPTAC used for proteogenomic analysis especially integration of signaling pathways and biomarkers discovery among large data set derived from RNAseq and proteomics analyses.

Comments on the Quality of English Language

The English is very good.

Just recheck minor spelling.

Author Response

The manuscript of Rui Vitorino presents a comprehensive review of the current high-throughput omics pipelines, covering key technologies like proteomics, peptidomics, metabolomics, next generation sequencing and gene expression, protein/DNA/RNA arrays coupled with data integration techniques and their diverse applications in the basic research and the biomedical field. 

In addition, the review presents the critical role of bioinformatics tools, clearly presented in the table 6 and highlights the major statistical methods in managing the large datasets generated by these technologies.

I recommend the review to be published in the Proteomes journal after minor revisions:

  1. The graphic abstract and the graph/scheme presented in the Figure 1 should be changed by using Arial font and larger size (for Figure 1 especially which is hard to be analyzed in the format presented in this version of the manuscript).

R: Thanks for your kind suggestion. I have revised and replaced by improved figures.

  1. Please add additional databases like CPTAC used for proteogenomic analysis especially integration of signaling pathways and biomarkers discovery among large data set derived from RNAseq and proteomics analyses.

R: Thanks for your kind suggestion. I have revised and added

“In the field of proteogenomic analysis, several databases have proven to be key resources, especially for pathway integration and biomarker discovery from large datasets derived from RNAseq and proteomic analyzes. The Clinical Proteomic Tumor Analysis Consortium (CPTAC) stands out as a key player in this area [51]. CPTAC offers an extensive repository of proteomic and genomic data across multiple cancer types, providing a rich resource for researchers looking to integrate multi-omics data. By utilizing CPTAC data, researchers can identify potential biomarkers and therapeutic targets by correlating protein expression profiles with genomic alterations. This comprehensive dataset enables the exploration of signalling pathways involved in cancer and facilitates the development of more targeted and effective therapies. The integration of proteomic data with genomic and transcriptomic information through CPTAC has significantly improved our understanding of cancer biology and treatment response [51]

One example of the integration of high-throughput genomic data is the analysis of colorectal cancer through proteogenomic characterization. Researchers integrated proteomic and genomic data from The Cancer Genome Atlas (TCGA) to identify novel proteomic subtypes of colorectal cancer. This study demonstrated that proteomic data provide additional insights beyond what genomic data alone can offer, including the identification of new molecular subtypes and potential therapeutic targets. The integration of these datasets highlighted specific proteins and phosphorylation events that were not apparent from genomic data, leading to a more comprehensive understanding of the disease and enabling more precise therapeutic interventions [55]. For example, in a proteogenomic study of a colorectal cancer cohort, a comparative proteomic and phosphoproteomic analysis of paired tumor and adjacent normal tissues was performed. This analysis provided a comprehensive catalog of colorectal cancer-associated proteins and phosphosites and identified both known and novel biomarkers, drug targets and cancer/testis antigens  [56]. The integration of proteogenomics data prioritized genome- derived targets, such as copy number drivers and mutation-derived neoantigens, providing new insights. In particular, phosphoproteomics data linked Rb phosphorylation to increased proliferation and decreased apoptosis in colorectal cancer, suggesting targeting Rb phosphorylation as a therapeutic strategy. In addition, proteomics showed that decreased CD8 T cell infiltration correlated with increased glycolysis in microsatellite instability-high (MSI-H) tumours, suggesting that glycolysis is a potential target to improve the efficacy of immune checkpoint blockade in MSI-H tumours. This study demonstrates how proteogenomics can reveal new biological insights and therapeutic opportunities [56].

Another remarkable resource is the Human Protein Atlas, which maps protein expression in various tissues and organs. This database is invaluable for integrating protein expression data with other omics datasets, such as RNAseq, to identify tissue-specific biomarkers. Similarly, the PhosphoSitePlus database [57] provides detailed information on post-translational modifications, such as phosphorylation sites, which are critical for understanding signaling pathways and regulatory mechanisms. ProteomicsDB [58] provides a comprehensive platform for human proteomics data, enabling the integration of proteomics with genomics and metabolomics data to uncover functional insights and interactions between metabolic pathways. In addition, the PRIDE (Proteomics Identifications Database) [59] repository supports the submission and retrieval of mass spectrometry-based proteomics data and promotes data sharing and integrative analysis. Together, these databases enhance the capacity for integrative multi-omics research by driving discoveries in biomarker identification and pathway elucidation, advancing the field of precision medicine.”

Reviewer 2 Report

Comments and Suggestions for Authors

This review presents discussion on omics data integration. The topic is too wide, so, the review covers very general aspects of high-throughput omics pipelines.

Strong points of this review of comparison of existing computer pipelines and tools.

The review is too general. It’s better to focus on proteomics data and pipelines.

Or show integration aspects – how to compare diverse data, existing tools, their advantages and disadvantages.

Figures and schemes are not logical. Need rearrange, at least make font size larger, make it readable.

Description of Ensembl is too common. There are other genome browsers with complex data processing worthy to mention (UCSF, IGV).

It is worthy add direct links to online tools to the text in addition to literature references (for example,  https://www.omicsnet.ca/ for OmicsNet)

Graphical Abstract should be updated.

Funny fonts and colors are not a problem. But way of connections of different blocks (ovals) should have some sense. Proteomics and metabolomics blocks should be  close to each other. Transcriptomics could be close to genomics, GWAS, and epigenetics.

Yellow oval (genomics and sequencing) could be close to each other.

Why is Network Analysis block shown separately?  

Network approach is a connecting element.

Please rearrange the ‘omics’ in the Graphical Abstract picture.

Line 68: “Tools such as Omics Integrator and G-language Genome” – need add references here.

Line 99: Figure 1. It is not visible! Please rearrange the blocks, make it larger in rectangular form, take more space (whole page, if necessary). The colors could be consistent (say one color for genomics, another for proteomics approaches). Otherwise just remove the colors from the figure. Currently, figure 1 is not satisfactory.

Table 1. Column ‘Needs’ should be renamed. It could be named as ‘Approaches’ or ‘Data sources’

Some words and phrases describing the tools are too common.

“offers powerful features” , “make it accessible to researchers”, “researchers can gain a deeper understanding”. Such phrases could be related to any tool.

Need compare the tools, show some practical suggestions how to use it better.

Please make text shorter removing redundant phrases. The paper will be only better without loss of information.

Line 166: “Tools such as CRISPR-Cas9 have also enabled…” – here the text is about molecular biology, please write it explicitly. Or is it a computer tool?

Table 2 is interesting, have the citation. But the title seems be misleading – “Concept And Need”.

It is not about ‘need’. Maybe word ‘problem’ suits here.

Line 228: ‘G-language Genome Analysis Environment’ – need a reference her

Line 229: ‘anvi’o’ – what is it? Tool name?

Line 257: ‘is integrating and analyzing…’ – I believe word ‘analyzing’ should be in another sentence. Just ‘integrating … the datasets’ has sense in this sentence.

Line 280: “exploring uncharted biological territory” – too literal. Please use direct language.

Line 344: “Modern bioinformatics platforms such as iProClass and iProXpress” – avoid word ‘modern’. All the bioinformatics tools are modern.

Add references after the tool name (iProClass), describe it a little.

Line 346: “perform comprehensive analyses and gain meaningful biological insights” – too complex wording, write shorter, direct phrase. It is just suggestion.

The text looks redundant due to long common phrases.

Line 356: ‘For example, tools such as BeFree’ – if write about ‘tools’ – need show at least two (or many different tools). Wording ‘such’ assume other known tools. But it is not shown in the text

Line 358 – NLP – give abbreviation in full.

Line 360: “PubTator tool” – need add  reference.

Line 392: ‘To this end, a bibliometric analysis’ – it is not clear – who did the analysis – the author, or it is reference to previous stufy?

“These articles were then analyzed…” – please add reference to ‘these articles’. I see no citations.

‘VOSviewer’ – add reference, URL to this tool.

Line 396: ‘in the DisGeNET’ – add reference to DisGeNet (be consistent in Upper-lower case letter writing – see DisGENET in Line 404)

Line 407: ‘Gene Expression Omnibus (GEO)’ – need reference, web-link.

Line 469: “Tools such as GenPipes facilitate” – add reference for GenPipes. Not use wording ‘tools such as’ when describing only one tool.

Line 512: ‘In another study,’ – change wording, give reference by number [].

Line 525: ‘Another application’ – it is again text repeat, word ‘Another’. It is rather ‘challenge for bioinformatics’ then an application.

Line 541: “The integration of genetic variants, DNA methylation and gene expression data 541

in bladder cancer research.” – looks like section title. Breal the paragraph here, at least.

Line 560: “National Human Genome Research Institute (NHGRI) and the European Bioinformatics Institute (EBI)” – add references to these platforms.

Line 568: “GWAS Integrator and the Phenotype-Genotype Integrator (PheGenI)” – add reference to these tools.

Line 622: ‘GKnowMTes’ -add reference

Table 4 has no reference in the text. Add a table note, what one can see there?

Line 676: DIA data  - wrong term (see DIA in line 673). Fix the phrase.

Line 677: ‘Open- SwathWorkflow and PyProphet’ – add references.

Line 686: ‘DIA-NN’ – give the abbreviation in full, add  the reference. Assume it is [82], but it shown on next page only. Add URL, web-links to online tools there it is appropriate in addition to literature references.

Table 5 seems have no text reference. How it appeared, it is author’s opinion only?

Line 727: “The development of sophisticated software platforms, such as the High-Throughput Autonomous Proteomic Pipeline (HTAPP)…” – again need reference here. Not use plural form (platforms) when describing only one platform.

Line 776: “Liver tissue, HepG2 cells and plasma are the most important areas.” – the phrase is not complete. Important areas there?

Line 781: ‘In this study’ – this or previous study without reference?

Table 6   - What data it is based on?

This table is good. I only suggest remove word Interoperability in ‘Interoperability Tools and Standards’ cell.

It is worthy to add reference (for Ensembl, KEGG, UniProt and other databases mentioned)

If no place in the table cells, make a table note.

Last line in the Table is a copy of header line. Is it typo?

It is important to cite all the databases and tools in the review!

Line 877: “OMIM, DisGeNET, UniProt, NCBI gene info, IID, MONDO, DrugBank, Reactome and DrugCentral” – give  all the references to these databases.

Line 881: “NeDRexApp, NeDRexAPI and the Neo4j” – references!

Line 883-885: “state-of-the-art network algorithms such as Multi-Steiner Trees (MuST), TrustRank…” – references!

Line 898: “A recent study” – give reference to this recent study!

Line 936: “MOMIC and OmicsSuite” – references!

Line 972: ‘Authors Contributions:’ – looks redundant. But might be formally it is accepted..

Reference 2( for Galaxy) is not complete (no volume, page numbers)

Ref. 108 – no page numbers.

Overall, this review has large number of references (111), but the references for important omics  tools are not shown in the main text.

Author Response

This review presents discussion on omics data integration. The topic is too wide, so, the review covers very general aspects of high-throughput omics pipelines.

Strong points of this review of comparison of existing computer pipelines and tools.

The review is too general. It’s better to focus on proteomics data and pipelines.

Or show integration aspects – how to compare diverse data, existing tools, their advantages and disadvantages.

R: Thank you for the valuable feedback. I appreciate the reviewer's suggestion to focus more specifically on proteomics data and pipelines, as well as the integration aspects of diverse data. While I understand the importance of honing in on proteomics, my intent was to provide a comprehensive overview that highlights the broader needs and challenges in the field of omics.

I have included a more examination of how various omics data can be integrated on table 1

I think that we need emphasize the necessity for developing user-friendly tools that do not require advanced programming skills. Highlighting the demand for accessible bioinformatics solutions will underline the importance of creating tools that cater to a broader audience, including researchers without extensive computational backgrounds. These enhancements will aim to balance the comprehensive nature of my review while providing specific insights into proteomics and data integration.

Figures and schemes are not logical. Need rearrange, at least make font size larger, make it readable.

R: Thanks for your kind suggestion. I have revised and replaced by improved figures.

Description of Ensembl is too common. There are other genome browsers with complex data processing worthy to mention (UCSF, IGV).

It is worthy add direct links to online tools to the text in addition to literature references (for example,  https://www.omicsnet.ca/ for OmicsNet)

R: Thanks for your kind suggestion. I have revised and included the respective references.

Graphical Abstract should be updated.

Funny fonts and colors are not a problem. But way of connections of different blocks (ovals) should have some sense. Proteomics and metabolomics blocks should be  close to each other. Transcriptomics could be close to genomics, GWAS, and epigenetics.

Yellow oval (genomics and sequencing) could be close to each other.

Why is Network Analysis block shown separately?  

Network approach is a connecting element.

Please rearrange the ‘omics’ in the Graphical Abstract picture.

R: Thanks for your kind suggestion. I have revised and replaced by improved figures following your suggestion.

Line 68: “Tools such as Omics Integrator and G-language Genome” – need add references here.

R: Thanks for your kind suggestion. I have revised and included the respective references.

Line 99: Figure 1. It is not visible! Please rearrange the blocks, make it larger in rectangular form, take more space (whole page, if necessary). The colors could be consistent (say one color for genomics, another for proteomics approaches). Otherwise just remove the colors from the figure. Currently, figure 1 is not satisfactory.

R: Thanks for your kind suggestion. I have revised and replaced by improved figures.

Table 1. Column ‘Needs’ should be renamed. It could be named as ‘Approaches’ or ‘Data sources’

Some words and phrases describing the tools are too common.

“offers powerful features” , “make it accessible to researchers”, “researchers can gain a deeper understanding”. Such phrases could be related to any tool.

Need compare the tools, show some practical suggestions how to use it better.

Please make text shorter removing redundant phrases. The paper will be only better without loss of information.

R: Thanks for your kind suggestion. I have revised and replaced by your suggestion.

Line 166: “Tools such as CRISPR-Cas9 have also enabled…” – here the text is about molecular biology, please write it explicitly. Or is it a computer tool?

R: I apologize for this misunderstanding. I rewrite this sentence:

“The advent of tools such as CRISPR-Cas9 has enabled precise genetic modifications, thereby facilitating functional studies on specific genes. Furthermore, high-throughput transcriptomics via RNA sequencing (RNA-seq) provides valuable insights into gene expression patterns under diverse conditions, assisting in the elucidation of gene regulatory networks and their roles in health and disease.”

Table 2 is interesting, have the citation. But the title seems be misleading – “Concept And Need”.

It is not about ‘need’. Maybe word ‘problem’ suits here.

R: I apologize for this misunderstanding. I rewrite the title.

Line 228: ‘G-language Genome Analysis Environment’ – need a reference her

Line 229: ‘anvi’o’ – what is it? Tool name?

R: Thanks for your kind suggestion. I have revised and included the respective references of these tools.

Line 257: ‘is integrating and analyzing…’ – I believe word ‘analyzing’ should be in another sentence. Just ‘integrating … the datasets’ has sense in this sentence.

R: Thanks for your kind suggestion. I have revised and replaced:

“One of the biggest challenges is integrating and interpreting the vast and heterogeneous datasets that these technologies generate. The complexity and scale of omics data require sophisticated computational methods and bioinformatics infrastructures to effectively process and understand the information.”

Line 280: “exploring uncharted biological territory” – too literal. Please use direct language.

R: I apologize for this misunderstanding.

“Over the past decade, omics technology has undergone significant advancements, evolving from its initial focus on cataloguing genes, proteins, and SNPs to performing disease-specific, in-depth analyses of various aspects of genomics, including meta-genetics, protein-protein interactions, modifications, and pathway mapping. Large-scale genome-wide association studies and high-throughput techniques have become more efficient and productive in exploring previously uncharted biological”

Line 344: “Modern bioinformatics platforms such as iProClass and iProXpress” – avoid word ‘modern’. All the bioinformatics tools are modern.

R: I apologize for this, I removed.

Add references after the tool name (iProClass), describe it a little.

R: Thanks for your kind suggestion. I have revised.

Line 346: “perform comprehensive analyses and gain meaningful biological insights” – too complex wording, write shorter, direct phrase. It is just suggestion.

R: Thanks for your kind suggestion. I have revised follow your suggestion.

The text looks redundant due to long common phrases.

Line 356: ‘For example, tools such as BeFree’ – if write about ‘tools’ – need show at least two (or many different tools). Wording ‘such’ assume other known tools. But it is not shown in the text

Line 358 – NLP – give abbreviation in full.

Line 360: “PubTator tool” – need add  reference.

R: Thanks for your kind suggestion. I have revised follow your suggestion.

Line 392: ‘To this end, a bibliometric analysis’ – it is not clear – who did the analysis – the author, or it is reference to previous stufy?

“These articles were then analyzed…” – please add reference to ‘these articles’. I see no citations.

‘VOSviewer’ – add reference, URL to this tool.

Line 396: ‘in the DisGeNET’ – add reference to DisGeNet (be consistent in Upper-lower case letter writing – see DisGENET in Line 404)

Line 407: ‘Gene Expression Omnibus (GEO)’ – need reference, web-link.

Line 469: “Tools such as GenPipes facilitate” – add reference for GenPipes. Not use wording ‘tools such as’ when describing only one tool.

Line 512: ‘In another study,’ – change wording, give reference by number [].

Line 525: ‘Another application’ – it is again text repeat, word ‘Another’. It is rather ‘challenge for bioinformatics’ then an application.

R: I apologize for this misunderstanding. I have revised follow your suggestion.

Line 541: “The integration of genetic variants, DNA methylation and gene expression data 541

in bladder cancer research.” – looks like section title. Breal the paragraph here, at least.

R: I apologize for this misunderstanding.

Line 560: “National Human Genome Research Institute (NHGRI) and the European Bioinformatics Institute (EBI)” – add references to these platforms.

Line 568: “GWAS Integrator and the Phenotype-Genotype Integrator (PheGenI)” – add reference to these tools.

Line 622: ‘GKnowMTes’ -add reference

R: I have include the respective references.

Table 4 has no reference in the text. Add a table note, what one can see there?

Line 676: DIA data  - wrong term (see DIA in line 673). Fix the phrase.

Line 677: ‘Open- SwathWorkflow and PyProphet’ – add references.

Line 686: ‘DIA-NN’ – give the abbreviation in full, add  the reference. Assume it is [82], but it shown on next page only. Add URL, web-links to online tools there it is appropriate in addition to literature references.

Table 5 seems have no text reference. How it appeared, it is author’s opinion only?

R: The table was constructed using comprehensive information gathered from a wide array of literature sources and references (please see below), in addition to my extensive experience in the field. This approach ensures that the table is both thorough and reliable, reflecting the current state of knowledge and practical insights in the subject area.

Line 727: “The development of sophisticated software platforms, such as the High-Throughput Autonomous Proteomic Pipeline (HTAPP)…” – again need reference here. Not use plural form (platforms) when describing only one platform.

R: I apologize for this misunderstanding.

Line 776: “Liver tissue, HepG2 cells and plasma are the most important areas.” – the phrase is not complete. Important areas there?

R: I apologize for this misunderstanding.

Line 781: ‘In this study’ – this or previous study without reference?

R: yes, I have include the reference.

Table 6   - What data it is based on?

This table is good. I only suggest remove word Interoperability in ‘Interoperability Tools and Standards’ cell.

It is worthy to add reference (for Ensembl, KEGG, UniProt and other databases mentioned)

If no place in the table cells, make a table note.

Last line in the Table is a copy of header line. Is it typo?

R: I apologize for this misunderstanding.

It is important to cite all the databases and tools in the review!

Line 877: “OMIM, DisGeNET, UniProt, NCBI gene info, IID, MONDO, DrugBank, Reactome and DrugCentral” – give  all the references to these databases.

Line 881: “NeDRexApp, NeDRexAPI and the Neo4j” – references!

Line 883-885: “state-of-the-art network algorithms such as Multi-Steiner Trees (MuST), TrustRank…” – references!

Line 898: “A recent study” – give reference to this recent study!

Line 936: “MOMIC and OmicsSuite” – references!

R: I have revised and included the respective references of these tools.

Line 972: ‘Authors Contributions:’ – looks redundant. But might be formally it is accepted..

Reference 2( for Galaxy) is not complete (no volume, page numbers)

Ref. 108 – no page numbers.

R: I apologize, it was endnote annotation. ´

Reviewer 3 Report

Comments and Suggestions for Authors

The review is informative, but the structure needs significant improvement.

General:

  • Please cite each table and figure in the main text.
  • The font size of Figure 2 is too small to read, and the font style is difficult to read.
  • The graphical abstract is misleading. The largest circle is labeled ‘applications of AI in omics,’ but AI is not the main topic of the review; machine learning is just a tool for omics integration. It would be better to put ‘integrative omics’ in the middle, as this is the main topic of the review.

1. Introduction (Lines 34-74):

This section is not general enough and focuses too much on specific tools. While it is good to provide information about tools, it would be better to have a separate section for them and list them in a table. Lines 34-78 can be removed. Lines 75-108 are a sufficient beginning.

1.3 Concept and Need (Lines 144-207):

The concept part of this section is repetitive with section 1.4, and the need part is repetitive with section 1.1. It would be more logical to move this section before section 1.2. First, discuss what each omics is (section 1.3 concept and section 1.4), then talk about why we need integration (section 1.1 and section 1.3 need) and how we can achieve that (section 1.2).

2. Case Studies and Applications (Lines 431-797):

  • Lines 364-382 would be better presented as a table.
  • Since the whole review is about omics integration, please elaborate on how these sections are related to omics integration. Titles of these sections seems off-topic. When providing examples, please clearly state which omics techniques were integrated, what disease questions were addressed, and what integration applications were used.

Author Response

The review is informative, but the structure needs significant improvement.

General:

  • Please cite each table and figure in the main text.
  • The font size of Figure 2 is too small to read, and the font style is difficult to read.
  • The graphical abstract is misleading. The largest circle is labeled ‘applications of AI in omics,’ but AI is not the main topic of the review; machine learning is just a tool for omics integration. It would be better to put ‘integrative omics’ in the middle, as this is the main topic of the review.

R: Thanks for your kind suggestion. I have revised and replaced by improved figures.

  1. Introduction (Lines 34-74):

This section is not general enough and focuses too much on specific tools. While it is good to provide information about tools, it would be better to have a separate section for them and list them in a table. Lines 34-78 can be removed. Lines 75-108 are a sufficient beginning.

R: I appreciate  your kind suggestion, which was followed.

1.3 Concept and Need (Lines 144-207):

The concept part of this section is repetitive with section 1.4, and the need part is repetitive with section 1.1. It would be more logical to move this section before section 1.2. First, discuss what each omics is (section 1.3 concept and section 1.4), then talk about why we need integration (section 1.1 and section 1.3 need) and how we can achieve that (section 1.2).

R: I appreciate  your kind suggestion, which was followed.

  1. Case Studies and Applications (Lines 431-797):
  • Lines 364-382 would be better presented as a table.

R: I appreciate  your kind suggestion, which was followed.

  • Since the whole review is about omics integration, please elaborate on how these sections are related to omics integration. Titles of these sections seems off-topic. When providing examples, please clearly state which omics techniques were integrated, what disease questions were addressed, and what integration applications were used.

R: I appreciate  your kind suggestion, which was followed at the end of introduction.

“The integration of high-throughput omics combines data from different omics technologies to gain a comprehensive understanding of biological systems. This integration is essential to unravel the complexity of cellular processes and disease mechanisms. The review explores advanced technologies and computational methods that facilitate omics integration, covering platforms such as next-generation sequencing (NGS) for genomics, RNA sequencing (RNA-Seq) for transcriptomics, mass spectrometry for proteomics, and nuclear magnetic resonance (NMR) spectroscopy for metabolomics. It addresses challenges such as heterogeneity, scale and standardization of data and proposes solutions such as advanced bioinformatics tools and machine learning techniques. Key applications include automated text mining techniques such as natural language processing (NLP) to extract meaningful information from scientific literature and genomic analyzes to identify biomarkers for disease to improve diagnostic tools and personalized medicine. Integrating data from resources such as the GWAS catalog helps identify genetic variants associated with different traits, supporting biomarker discovery and therapeutic targets. Proteomics, facilitated by mass spectrometry, provides insights into protein functions and interactions, and the integration of proteomics data with other omics datasets improves the understanding of disease mechanisms. Effective integration strategies, such as early, mixed, middle, late and hierarchical integration, are essential for comprehensive insights into complex biological systems.”

Reviewer 4 Report

Comments and Suggestions for Authors

The review paper presented by Vitorino et al summarized current research efforts on integrating high-throughput omics data, particularly the proteomics and other omics data types. This is a very comprehensive review on the relevant topics. I found it helpful to the general readers interested in omics integrations. While the main discussions are very well presented, I have only have a few suggested edits detailed below.

11. In addition to summarizing the functions of existing tools, it is also beneficial to inform the readers what are limitations of current tools and methods. Some discussions on the possible solutions for these challenges could also be helpful in each subsection. These contents need not to be lengthy in-depth discussions, but a short discussion for each subsection should be sufficient.

22.      Can the authors introduce the basic computational principles behind various integration methods? E.g. whether the methods are focusing on finding the similarity among omics modalities or differences? A brief introduction on how the most popular methods work could also be helpful. This could be some be high-level overview of the algorithm details.

33.      The authors may modify the title to suggest the clinical implications in the manuscript, since this has been mentioned many times.

Comments on the Quality of English Language

English is well-written and understandable

Author Response

The review paper presented by Vitorino et al summarized current research efforts on integrating high-throughput omics data, particularly the proteomics and other omics data types. This is a very comprehensive review on the relevant topics. I found it helpful to the general readers interested in omics integrations. While the main discussions are very well presented, I have only have a few suggested edits detailed below.

  1. In addition to summarizing the functions of existing tools, it is also beneficial to inform the readers what are limitations of current tools and methods. Some discussions on the possible solutions for these challenges could also be helpful in each subsection. These contents need not to be lengthy in-depth discussions, but a short discussion for each subsection should be sufficient.

R: Thanks for your kind suggestion. Following the similar feedback of the R2, table was constructed using comprehensive information gathered from a wide array of literature sources and references.

  1. Can the authors introduce the basic computational principles behind various integration methods? E.g. whether the methods are focusing on finding the similarity among omics modalities or differences? A brief introduction on how the most popular methods work could also be helpful. This could be some be high-level overview of the algorithm details.

R: Thanks for your kind suggestion. Following it:

“1.        Introduction

The Omics high-throughput pipeline is transforming biological research by enabling comprehensive, large-scale analysis of diverse biomolecular data. These advanced technologies generate extensive data at multiple omics levels, including genomics, transcriptomics, proteomics and metabolomics. To handle the complexity and volume of this data, sophisticated bioinformatics pipelines are required that integrate various software tools and databases to pre-process, analyze and interpret the data, forming intricate workflows. The integration of high-throughput omics is primarily based on two fundamental approaches: similarity-based methods and difference-based methods. Similarity-based methods aim to identify common patterns, correlations and common paths in different omics datasets. These methods are crucial for understanding overarching biological processes and identifying universal biomarkers. For example, correlation analysis evaluates the correlation between different omics levels (e.g. genomics, transcriptomics, proteomics) to find common trends and relationships and identify co-expressed genes or proteins in different datasets. Clustering algorithms such as hierarchical clustering and k-means clustering group similar data points from different omics datasets and uncover modules or networks of genes and proteins that work together. Network-based approaches such as Similarity Network Fusion (SNF) construct similarity networks for each omics type and then integrate them into a single network, merging information from all omics levels to highlight commonalities and identify important biological pathways. On the other hand, difference-based methods focus on detecting unique features and variations between different omics levels, which is essential for understanding disease-specific mechanisms and for personalized medicine. Differential expression analysis compares the expression levels of genes or proteins between different states (e.g. healthy vs. diseased) to identify significant changes and recognize unique molecular signatures associated with specific conditions [1]. Variance decomposition decomposes the total variance observed in the data into components attributable to the different omics levels. This helps to understand the contribution of each omics type to the overall variability and to identify omics-specific variation. Feature selection methods such as LASSO (Least Absolute Shrinkage and Selection Operator) and Random Forests select the most relevant features from each omics dataset and integrate these features into a comprehensive model that captures the unique aspects of each layer [2]. Popular integration algorithms include Multi-Omics Factor Analysis (MOFA) and Canonical Correlation Analysis (CCA). MOFA is an unsupervised approach that uses Bayesian factor analysis to identify latent factors responsible for variation in multiple omics datasets and integrates the data to identify underlying biological signals. CCA identifies linear relationships between two or more omics datasets, facilitating the discovery of correlated traits and common pathways [3].

For example, in genome analysis, the entire set of genes in an organism's genome is analyzed. Tools such as Ensembl (https://www.ensembl.org/) and Galaxy [4] are useful for this purpose. a web-based open source platform with a user-friendly interface for bioinformatics analysis workflows.”

  1. The authors may modify the title to suggest the clinical implications in the manuscript, since this has been mentioned many times.

R: Thanks for your kind suggestion. Following it:

“Transforming Clinical Research: The Power of High-Throughput Omics Integration”

Round 2

Reviewer 2 Report

Comments and Suggestions for Authors

Thanks for the revision and detailed answer. I have no more critical remarks.

Author Response

Thank you.

Reviewer 3 Report

Comments and Suggestions for Authors

The revised version is well-organized. Those tables are informative. 

I do not have further comments. 

Author Response

Thank you.